# Canine detection of explosives under adverse environmental conditions with and without acclimation training

**Sarah A. Kane**[1]*, **Lauren S. Fernandez**[1], **Dillon E. Huff**[2], **Paola A. Prada-Tiedemann**[2], **Nathaniel J. Hall**[1]

**1** Canine Olfaction Research and Education Lab, Davis College of Animal and Food Science, Texas Tech University, Lubbock, TX, United States of America, **2** Forensic Analytical Chemistry and Odor Profiling Laboratory, Department of Environmental Toxicology at Texas Tech University, Lubbock, TX, United States of America

* sarahkan@ttu.edu

**Data Availability Statement:** All relevant data are within the manuscript and its Supporting Information files.

## Abstract

Canines are one of the best biological detectors of energetic materials available; however, canine detection of explosives is impacted by a number of factors, including environmental conditions. The objectives of this study were: 1) determine how canine detection limits vary when both the canine and odorant are tested in varying temperature and humidity conditions (canine and odor interactive effects); and 2) determine if an acclimatization plan can improve detection limits in an adverse environmental condition. Eight working line canines were trained to detect four energetics: prill ammonium nitrate (AN), Composition 4 (C4), tri-nitrotoluene (TNT) and double base smokeless powder (SP). In Experiment 1, canines completed a 3-alternative forced choice 3-down-1-up staircase threshold assessment in five environmental conditions: 40°C and 70% relative humidity (RH), 40°C and 40% RH, 0°C and 90% RH, 0°C and 50% RH and 21°C and 50% RH. Canines showed a 3.5-fold detection limit increase (poorer detection) for C4 in 40°C and 70% RH compared to their detection limit at 21°C and 50% RH. In Experiment 2, the eight canines were split into two groups (n = 4), control and acclimation groups. The control group completed the threshold assessment for C4 at 21°C and 50% RH each day for 20 days, with 5 minutes of petting prior to testing. The acclimation group completed the same assessment daily starting at 21°C and 50% RH but temperature and RH were incremented daily over the course of 6 days to the 40°C and 70% RH condition. After the initial six days, the acclimation group completed daily assessments at 40°C and 70% RH condition for the remainder of the experiment. All acclimatization group canines started their session with 5 minutes of toy or food retrieves. Detection limits for C4 for all dogs were tested in 40°C and 70% RH on day 11 and day 22. The acclimatization plan improved detection limits in the 40°C and 70% RH condition for C4 compared to the non-acclimated group. In this set of experiments, canine detection limits for four explosive odorants were found to vary based on environmental condition and were mostly driven by impacts on the canine rather than odor availability. The acclimatization plan did result in lower detection limits (i.e., increased performance). Future work should determine what

**Funding:** This research was made possible through funding provided by the DoD Army Research Office under Contract No. W911NF2120124. The views expressed in this manuscript are those of the author and do not necessarily reflect the official policy or position of the Department of Defense, nor the U.S. Government. https://www.arl.army.mil/who-we-are/aro/ SAK's work was supported by the National Science Foundation Graduate Research Fellowship Program (DGE 2140745). Any opinions, findings, conclusions or recommendations expressed in this work are those of the author(s) and do not necessarily reflect the views of the National Science Foundation.https://www.nsf.gov/ The funders had no role in study design, data collection and analysis, decision to publish, or preparation of the manuscript.

**Competing interests:** The authors have declared that no competing interests exist.

factor (exercise or environmental exposure) is more effective in acclimatization for odor detection work.

## Introduction

Explosive detection canines (EDCs) are critical for public and military safety. Currently, specially trained canines are one of the best detectors for explosives because of their mobility, and capacity to deploy in a variety of environments [1]. EDCs are able to search environments which other detection technologies cannot access [1]. In addition, EDCs can be deployed in a variety of environments, both indoor and outdoor [2]. Farr et al. (2021) surveyed EDC handlers who reported they were regularly deployed to search in climates to which they had limited acclimation prior to searching.

Although the versatility of EDCs is a benefit, further research is needed to understand how temperature and humidity can affect their detection abilities [3]. Because EDCs can be rapidly deployed to harsh environments, it is crucial to know how these environmental changes can impact their detection limits. In addition, such research could provide a useful resource for handlers on best practices for working EDCs in adverse conditions.

Much of what we know about environmental effects on detection limits does not come from the EDC literature, but from conservation detection dogs. In one such study, relative humidity and air temperature were not correlated with detection rates of the desert tortoise [4]. This lack of a result could be procedural as canine teams were rested once the dog showed behavioral signs of heat stress, or their rectal temperature was greater than 40˚C. Because dogs were only worked when they were in optimal physiological conditions, fatigue or stress caused by the environment did not impact their detection rates. This study did note that canine fatigue was a limiting factor in duration of search [4]. It was unclear if that fatigue was due to the physical demands of the task, the environmental conditions in which the canines worked, or a combination of both. Another study, however, found that temperature did affect detection rates; higher air temperatures and higher windspeeds led to greater detection distances (better detection) [5]. Unfortunately, due to the lack of climactic variation in this study, further conclusions on the effect of climate (specifically relative humidity) on detection were limited.

Gazit and Terkel (2003) studied how physiological factors, specifically exercise induced fatigue, affected canine detection of explosives. Gazit and Terkel (2003) compared canines' ability to detect C4 in indoor and outdoor search areas under two conditions: after rest and after strenuous physical activity. They found that canines were always able to detect the explosive regardless of condition, in the indoor search, however after strenuous activity, canines missed explosives in their outdoor search. Search time was also increased, both indoor and outdoor, after exercise. Gazit and Terkel (2003) suggested that the decreased ability to detect C4 after activity was because of changes in the canines' sniffing to panting ratio. In order to maintain homeostasis during and after exercising, canines need to pant to cool via evaporative cooling [6]. Gazit and Terkel (2003) posited that as panting increased, sniffing deceased, and thus detection of C4 decreased. Canines did show acclimatization throughout the study to detect C4 after exercise [7].

Past research in our laboratory has studied how canine physiological factors alone affect detection limits of narcotic signatures and explosives (Brustkern et al., 2023; Fernandez et al., in review). Fernandez et al. (in review) worked canines through a go/no-go threshold assessment with four different explosive targets. The canines worked in an environmentally

controlled chamber; however, the odorant was kept outside of the research area in a temperature-controlled water bath. The purpose of this study was to determine how canine factors alone affected detection limits for explosives. This study found that canines had lowest detection limits in standard (22°C and 50% RH) conditions compared to environments with higher or lower temperatures and RH levels. A similar study which used methyl benzoate as the target material, also kept the odorant in a consistent environment while the dog worked in a variety of temperature and humidity conditions [8]. This study found that canines had poorer detection in high temperature conditions compared to standard, but did not see a performance decrement in low temperature conditions. This study also found that canines had poorer detection immediately upon entering the harsh environment from a controlled condition, i.e. there was a "startup" decrement [8].

In addition to canine factors, odor availability is also affected by the environment [1]. In high temperature and relative humidity conditions more volatiles of an explosive enter the headspace. This should allow for greater availability for canine detection. Conversely, colder conditions decrease volatiles in the headspace and lead to decreased odor availability [9]. Both odor and canine physiological state fluctuate with environmental conditions, and these interactions determine detection limits. Huff et al. (2022) studied how explosive odors behave in a variety of temperature and relative humidity conditions using SPME and GC-MS techniques. In general, when sampling occurred in hot conditions, volatiles in the headspace increased as expected and decreased in colder conditions (9).

It remains unclear, however, how odor availability and canine behavior interact when in adverse environmental conditions together. It is possible that any challenges dogs experience in high temperatures could be mitigated by increased odor availability in these temperatures. Conversely, it is possible that despite elevated temperatures, and increased odor availability, physiological effects on the canine may decrease detection, as seen by Gazit and Terkel (2003). In cold conditions, odor is less available, but this effect may be lessened if the dogs are not under physiological stress (i.e. panting) when working in this condition.

It is evident in the literature that there is a lack of conclusive support on how climate, odor and canine detection interact. The objective of the first experiment of this study was to determine how detection limits of canines are affected by environmental condition when both odorant and canine are held in the same environmental conditions. The second experiment objective was to develop an acclimatization plan to improve detection in adverse conditions.

## Experiment 1: Characterizing canine detection sensitivity in varying environments

### Materials and methods

**Participants.**   Eight dogs (6M, 2F; 4 Labrador retrievers, 4 German Short-Haired Pointers), former candidates of a government detection program, participated in this project. The dogs had various amounts of prior training with explosives.

**Animal welfare considerations.**   This project was conducted at Texas Tech University and was overseen by the Institutional Animal Care and Use Committee (Protocol #21051–07) and approved by the US Army Medical Research and Development Command Animal Care and Use Office (#78018-ST-H.e001). In addition to their participation in this project, all the dogs had additional enrichment activities every day and received food enrichment. Throughout training and testing, all dogs always had access to water. During testing, if canines refused to search for 5 trials in a row or laid down and did not get up when called, testing was stopped, and they were immediately removed from the environmental chamber. This termination

**Table 1. Explosive material used as target odorants in Experiment 1.** Vial diameter is specified because surface area impacts odor availability. Explosives held in a vial with a larger diameter had a higher odor availability than if they had been held in a vial with a smaller diameter. Larger diameter vials were used when odors were difficult to detect even at the initial standard condition.

| Odor | Mass per vial (g) | Brand of energetic material | Vial diameter (cm) |
|---|---|---|---|
| Double-base smokeless powder (SP) | 1.039g | Hodgdon H335 | 2.54 |
| Ammonium nitrate (AN) | 20.422g | Omni Explosives | 5.08 |
| Trinitrotoluene (TNT) | 20.6 | Omni Explosives | 5.08 |
| C4 | 4.23g | Omni Explosives | 2.54 |

criterion was used to ensure participant safety; this criterion never occurred under standard conditions and was indicative of lack of performance.

**Odorants.** Four odorants were used in this project; C4, double-based smokeless powder, ammonium nitrate prill (explosives grade), and TNT (see Table 1 for amounts and brands). Use of all energetic materials was approved by the Institutional Laboratory Safety Committee on Energetic Materials (Protocol ILSC#2110E1). There were no distractor odorants, all other valves of the olfactometer (described below) were connected to empty borosilicate glass vials matching the type used to contain the target odorants.

**Data collection equipment.** Three Bluetooth enabled olfactometers were used in this project, which were a modification of our prior automated line up equipment (see Aviles-Rosa et al., 2021). The odor generation design was identical, but instead of having a horizontal panel of odor ports, the odor port was integrated into the olfactometer box itself. Additionally, the programing was modified to allow Bluetooth control of the olfactometers.

Each olfactometer was an independent replicate containing the same odor delivery system and odorants. Each olfactometer had six vials: one vial contained the target odorant (C4, TNT, AN, or SP) while the remaining five vials were blank controls. These control vials were cleaned and prepared in a manner identical to the target vial.

Odor was generated from a 4L/minute oil-less air pump. Air from the pump first passed through a charcoal filter to scrub the air of odorants. The air was then split to an odor line and a dilution line via a T-connector. The odor line was regulated by a 0–1 lpm rotameter and the dilution line was regulated by a 0–4 lpm rotameter. From that regulating rotameter, the odor line passed to a manifold with one-way valves. Upon activation of a valve, the odor line flow was passed into a vial containing the odorant (or a clean distractor vial). This displaced the headspace of the vial through a one-way check valve into a PTFE mixing manifold where it mixed with clean air from the regulated clean air. Air from this manifold flowed to the port for the dog to sniff.

The olfactometer was controlled by a custom program (https://github.com/njhall1/CanineOlfactometer). Odor dilutions were controlled by manual manipulation of the flowmeters controlling the odor line and dilution line. Thus, dilution was created as an air dilution by changing the ratio of air from the odor and dilution lines. The flowmeters allowed a dilution from 80 to 3% (see Table 2). For threshold testing, a series of dilutions was designed: 80%, 50%, 25%, 12% and 3%. The flow rate, or the amount of odor being pushed into the port was consistent for each concentration, except for 80%. To reach to the 80% concentration, the flow rate had to be lower, however all three olfactometers delivered the same flow rate.

Each olfactometer contained an infrared sensor beam pair that measured all canine entries into the odor ports. These IRs also recorded when the beam was broken so that duration between nose pokes in each box could be calculated. We also calculated latency, or the duration from start of the trial to the time the dog began search (as indicated by a nose poke breaking the IR beam).

**Table 2. Odor concentration changes for the 3 down 1 up staircase threshold procedure.**

| Concentration (% odor line to clean air line) | Step number | Odor line flow rate (cc/min) | Clean line flow rate (L/min) |
|---|---|---|---|
| 80% | 1 | 1000 | 0.25 |
| 50% | 2 | 1000 | 1.0 |
| 25% | 3 | 750 | 2.25 |
| 12% | 4 | 360 | 2.65 |
| 3% | 5 | 90 | 2.91 |

This study utilized an alternative forced choice test design, in which the target odor was always present in one of the three olfactometers. The other two olfactometers were pushing the same volume of air from one of the five control (empty) vials.

**Temperature sensor.** Each dog had a subcutaneous microchip over their shoulder which read their subcutaneous body temperature. At the start of every testing session, approximately every 5 minutes during the session, at the end of the session and 5-minutes post session, an experimenter scanned the 134.2 kHz ISO microchip with a HomeAgain™ Universal Worldscan™ Reader Plus (by MERCK animal health) to record the subcutaneous temperature of the dog [10].

**Environmental conditions.** There were five temperature and humidity conditions used in this experiment (see Table 3). Experimenters allowed the standard, low-temperature high humidity, high temperature low humidity and high temperature high humidity to vary by 3˚C and 5% relative humidity (RH) from the stated condition to allow for natural fluctuations as airflow changed when the door of the environmental chamber was opened, or the AC unit began a thaw cycle. The low temperature low humidity condition was allowed to vary by 3˚C and 10% RH due to ambient humidity levels.

Dogs were tested once under each condition for each target odor. All dogs were first tested with each odor under standard conditions. Four dogs were randomly assigned to one of two testing orders (A or B; see Table 4 for order). Each odor was tested to completion before moving to the next to reduce the potential of olfactometer contamination by switching odors multiple times per day. Threshold limits were determined first for C4 then SP, AN, and last TNT.

**Experimental set up.** Research was conducted in a 3.6 x 3.4 m environmental chamber. The room was equipped with a heater, AC unit, dehumidifiers (hOmeLabs and ALORAIR), humidifiers (AILINKE), a fan and a mister system (homenote Misting Cooling System). The misters hung from the ceiling to help the room reach humidity conditions. The room also contained a SensorPush HT.w wireless thermometer/hygrometer sensor which measured the humidity and temperature in the room (rated accuracy: ± 2% RH and ± 0.2˚C). Three Bluetooth enabled olfactometer boxes were arranged as a triangle in the room and presented all odors for canines (Fig 1).

**Table 3. Temperature and humidity conditions for each of the 5 environmental conditions and the allowed for variation in the environmental condition set point.**

| Condition | Temperature (˚C) | Relative humidity (% RH) |
|---|---|---|
| Standard (STD) | 21 ± 3 | 50 ± 5 |
| High temperature high humidity (HTHH) | 40 ± 3 | 70 ± 5 |
| high temperature low humidity (HTLH) | 40 ± 3 | 40 ± 5 |
| Low temperature high humidity (LTHH) | 0 ± 3 | 90 ± 5 |
| Low temperature low humidity (LTLH) | 0 ± 3 | 50 ± 10 |

**Table 4. Indicates the orders each group of dogs worked through the conditions for each target odorant.** Blocking of testing order was done to help limit order effects. STD = standard, HTHH = high temperature high humidity, HTLH = high temperature low humidity, LTHH = Low temperature high humidity, LTLH = low temperature low humidity. Odors ordered in the table as they were presented to the canines, top to bottom. Groups were consistent throughout experiment 1.

| Groups (n = 4) | Odor | Cond 1 | Cond 2 | Cond 3 | Cond 4 | Cond 5 |
|---|---|---|---|---|---|---|
| Group A | C4 | STD | LTHH | HTHH | HTLH | LTLH |
| Group B | C4 | STD | HTLH | LTLH | LTHH | HTHH |
| Group A | SP | STD | HTLH | LTHH | LTLH | HTHH |
| Group B | SP | STD | LTHH | HTLH | HTHH | LTLH |
| Group A | AN | STD | HTHH | LTLH | LTHH | HTLH |
| Group B | AN | STD | LTHH | HTHH | LTLH | HTLH |
| Group A | TNT | STD | LTHH | HTHH | HTLH | LTLH |
| Group B | TNT | STD | HTHH | HTLH | LTLH | LTHH |

Throughout the duration of the experiment, dogs were kept in runs with continuous outdoor access. Dogs were walked twice daily for at least 20 min per walk for the duration of this study. The average high outdoor temperature during the duration of this experiment was 33.5˚C with average lows of 17.5˚C.

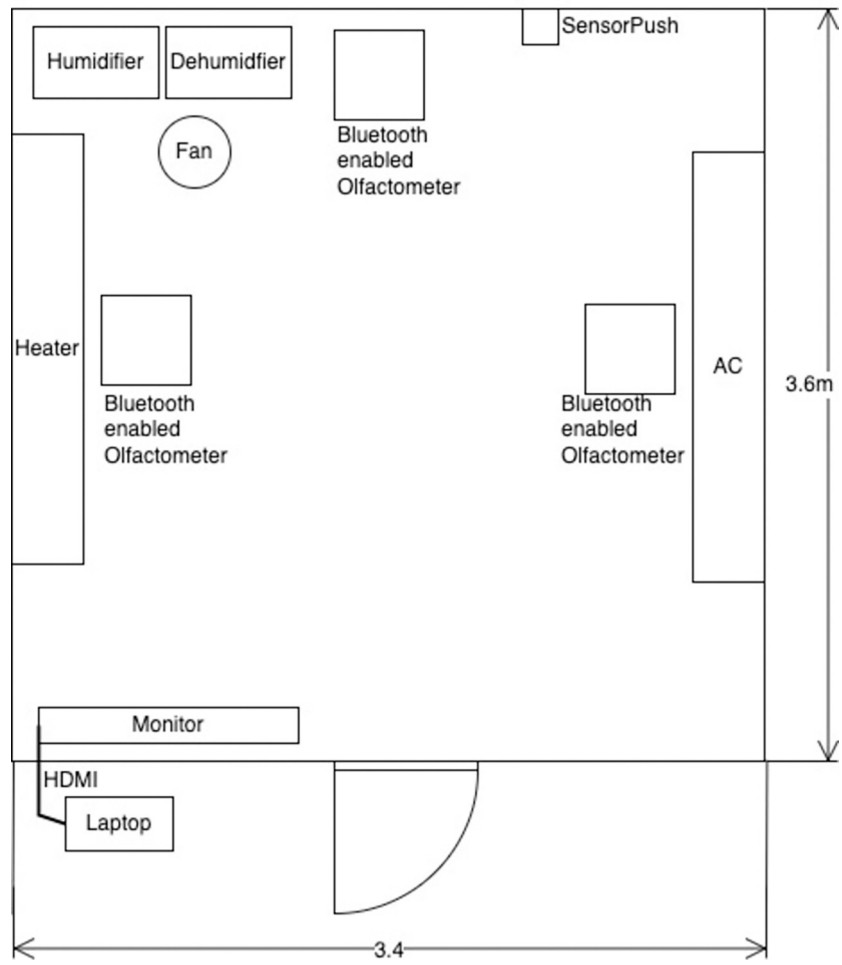

**Fig 1. Environmental chamber room set up (overhead view).** The humidifier and dehumidifier number varied based on conditions requirements. The Bluetooth enabled olfactometer boxes (3) or BLE olfactometers are pictured.

**Odor availability analysis.** A solid phase microextraction headspace sampling was performed on all target explosive odorants while being exposed to each extreme condition inside the environmental chamber. This extraction mode aimed to understand how the chemical odor profile remained during an "active operational" environmental. Each environmental condition and target odorant combination included a total of ten replicate samples.

All chromatographic separations utilized an Agilent 7890A GC Series GC system, equipped with front and rear split/splitless injection ports, fitted with an HP5 30 m, 0.25 mm internal diameter capillary column. Helium was the carrier gas at a flow rate of 1 mL min-1 at an average velocity of 37 cm s-1. The mass spectrometer (MS) operated in electron ionization (EI) full scan mode from 45 to 550 amu, with a 3-min solvent delay. All compounds in the headspace were assigned based on mass spectral matches to the NIST 2017 mass spectral library.

Quantification of target VOC concentrations being extracted by the SPME fiber was performed via external calibration using a standard mix of signature VOCs within each explosive class. For the target VOCs of each explosive class, the calibration curve range between 1–100 mg $L^{-1}$ to demonstrate the fiber capacity to extract target analytes at trace concentrations in addition to higher concentrations quantified above 100ppm without losing linearity as observed with the linear dynamic range coefficients of > 0.98. To approximate the amount of VOCs extracted by the SPME fiber, the slope of the line obtained in the calibration curve was used as a response factor for each of the analyzed compounds.

**Canine <u>odor</u> detection testing and training.** All canines had been previously trained to detect explosives in an olfactometer during a previous study [11]. Dogs were trained to detect the target odorants in a 3 alternative forced choice test paradigm prior to starting threshold testing. Canines all reached 85% accuracy at the 80% concentration across 20 trials for each odor before beginning threshold testing on that odor.

All training and testing was conducted under double blind conditions such that the handler was never aware of the location of the olfactometer presenting the target odor. Canine responses (i.e. alerts), were scored automatically using infrared beam pairs. All dogs had a nose hold and freeze alert response for the target odorants. If dogs made a correct response, a tone played, and the handler provided a reinforcer to the dog. If the dog made a false alert or failed to respond to the correct port, an "incorrect" tone played. When the dog was incorrect no reinforcer was given. At the end of the trial, there was a 20s inter-trial interval to purge the olfactometers of any residual odor.

**Canine odor threshold testing.** The threshold testing was conducted as an alternative forced choice task with a 3-down 1-up descending staircase procedure. In an alternative forced choice task, the target odorant was present in one of the three olfactometers in every trial. In the 3- down, 1- up threshold test procedure, after three consecutive correct trials the odor concentration was decreased by one dilution step. If the dog made an incorrect response, the odor concentration was increased by one step. A reversal point was defined as anytime the direction of odorant concentration changed. Threshold testing continued until dogs completed seven reversals. Testing also terminated if dogs made three consecutive correct responses at the lowest concentration step (3%). Additionally, testing also terminated if dogs met a welfare criterion by not searching for five consecutive trials or showed signs of elevated heat stress and refused to search for one trial. Elevated heat stress signals consisted of a dog lying down in the testing chamber, excessively panting, and refusing to continue searching for one trial. The criterion of trials not searching was reduced when accompanied by these additional signs of heat stress.

Dogs remained in the environmental chamber until they had completed testing by achieving 7 reversals, scoring 3 correct responses at 3% concentration, until they had been in the chamber for 40 minutes, or if they met our welfare criterion. Most testing sessions were approximately 25 minutes.

**Data analysis.** Odor detection threshold was calculated as the geometric mean of all seven reversal points. Threshold was log transformed for all analyses. If dogs' detection exceeded the maximum dilution capabilities of the olfactometer (3 correct responses at the lowest concentration), the lowest dilution point was used as a reversal point for the remaining reversal points of that session. For example, if a dog reached 3% by reversal 2, the remaining five reversal point concentrations were imputed as 0.03. If a dog terminated a session early due to meeting the welfare criterion, the remaining reversal points were imputed as 80%, indicating failure to complete the task.

To evaluate the effect of environmental conditions on detection limits, a linear mixed effect regression model was fit in which log threshold was predicted by breed, environmental condition, odor and an interaction between environmental condition and odor, with a random effect of individual dog. This model produced a residual plot with some boundary effects, due to the limitations of the highest and lowest concentrations produced by the olfactometers (maximum concentration was 80% and minimum was 3%). Otherwise, the residual plot showed no patterns and so the model was deemed fit for interpretation. Models were fit in R using the lme4 package [12] and significance interpretation was made using the Anova function from the car package [13] and lsmeans from the emmeans package [14].

## Results and discussion

The thresholds for each odor in each condition can be seen in Table 5 and Fig 2. The detection limits (log scale) for each odor varied by condition and odor identity, supporting previous work [11]. For smokeless powder, dogs' detection limits were lower than the dilutions available by the Bluetooth olfactometers. This was expected based on the detection limits observed by Fernandez (2023).

Post hoc tests were conducted comparing each environmental condition to standard conditions for each target odor (see Table 6). Fig 2 shows the average geometric means for all dogs in each condition for each odor and the 95% confidence intervals. Post hoc tests showed that detection limits for C4 were higher (poorer) under high temperature high humidity (est = 0.47, t = 2.7, p = 0.03) and high temperature low humidity conditions (est = 0.38, t = 2.3, p = 0.05). Low temperature low humidity and low temperature high humidity conditions did not differ from standard conditions (p>0.05). Smokeless powder thresholds did not differ between environmental conditions (all p>0.05), but this may reflect that dogs' thresholds was below the limits of dilution of the olfactometer.

**Table 5. Air dilution detection limit (on the log scale) by odor in each condition.**

| Odor | | Standard | High Temp High Humid | High Temp Low Humid | Low Temp High Humid | Low Temp Low Humid |
|---|---|---|---|---|---|---|
| AN | Mean | -0.73 | -0.35 | -0.45 | -0.45 | -0.90 |
| | 95% CI | -0.99, -0.47 | -0.60, -0.090 | -0.71, -0.20 | -0.70, -0.18 | -1.2, -0.65 |
| C4 | Mean | -1.4 | -0.95 | -1.03 | -1.2 | -1.39 |
| | 95% CI | -1.7, -1.2 | -1.2, -0.69 | -1.29, -0.77 | -1.5, -0.97 | -1.7, -1.1 |
| SP | Mean | -1.5 | -1.5 | -1.5 | -1.5 | -1.2 |
| | 95% CI | -1.8, -1.2 | -1.7, -1.2 | -1.8, -1.3 | -1.7, -1.2 | -1.5, -0.96 |
| TNT | Mean | -0.79 | -0.53 | -0.88 | -0.47 | -0.79 |
| | 95% CI | -1.05, -0.53 | -0.78, -0.27 | -1.14, -0.62 | -0.72, -0.21 | -1.05, -0.53 |

The linear mixed effect model indicated that there was a significant effect of odor ($X^2$ = 181 DF = 3, P<0.0001), environmental conditions ($X^2$ = 13.03, DF = 4, P = 0.01) and their interaction ($X^2$ = 26.4, DF = 12, P<0.01) on detection limit. Breed, however, had no impact ($X^2$ = 0.22, DF = 1, P = 0.64).

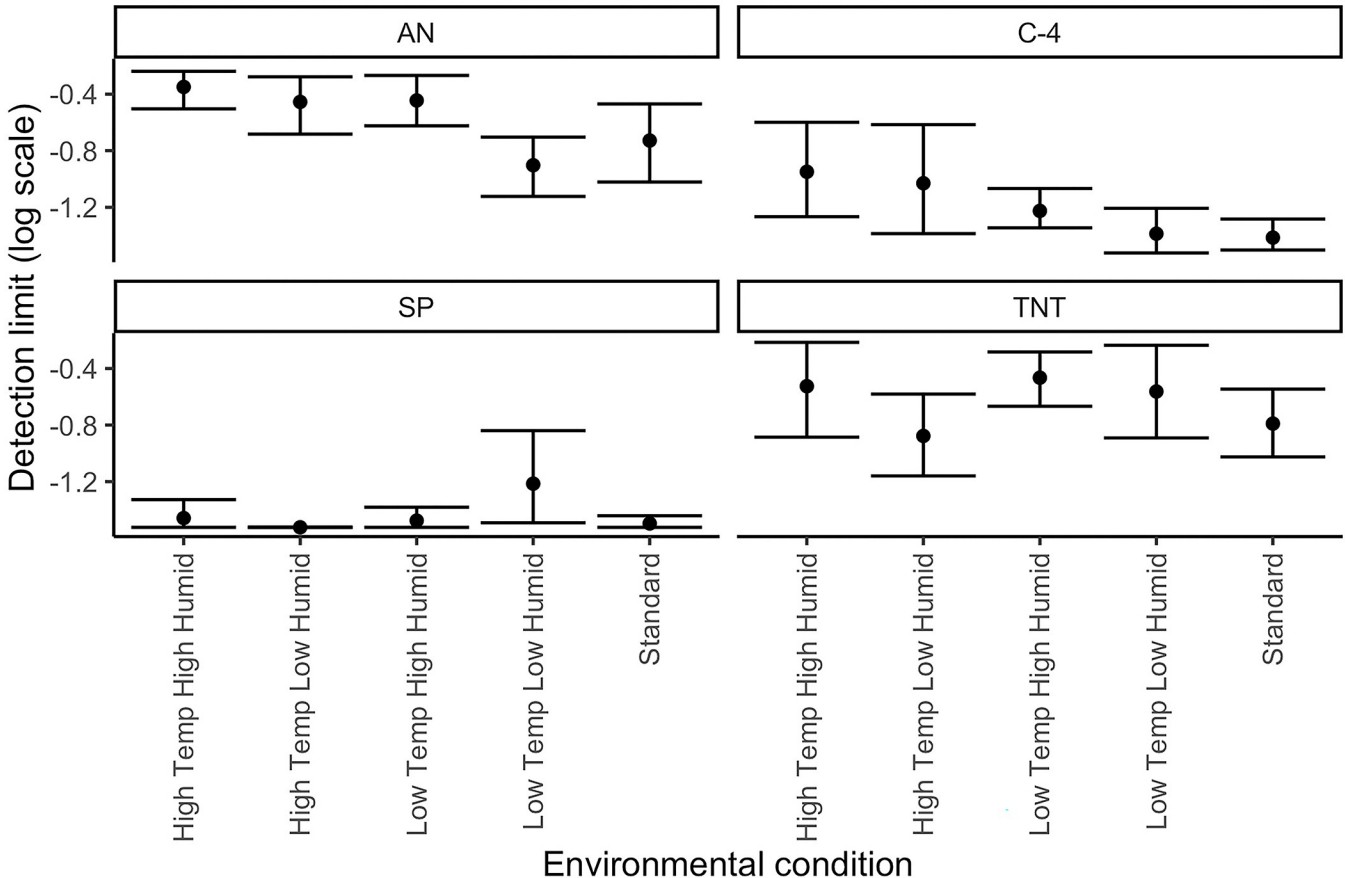

**Fig 2. Detection limit (log scale) for each odorant in each environmental condition.** The figure indicates that the detection limit for smokeless powder (SP) was not reached. The points on the graph are the average log geometric mean thresholds for all dogs in each condition for each odor. The error bars represent the 95% confidence interval for those geometric means.

For AN, poorest detection limits occurred under the hot temperature and low temperature high humidity conditions, however these differences from standard did not reach the statistical significance criterion (all p>0.05). For TNT, dogs showed detection difficulty in all conditions and no environmental condition differed from standard.

**Canine subcutaneous temperature and detection limit.** Fig 3 shows average canine subcutaneous temperature readings across sessions. Only the first 25 minutes are shown as most data are captured within the first 25 minutes of a session. Similar to that observed by Fernandez (2023), canine subcutaneous temperatures were highest in the high temperature and high humidity condition. A simple mixed effect model evaluated the impact of environmental

**Table 6. Detection limit comparisons for each odorant in each condition vs. standard.**

| Conditions | High Temp High Humid | | High Temp Low Humid | | Low Temp High Humid | | Low Temp Low Humid | |
|---|---|---|---|---|---|---|---|---|
| Odor: | t-ratio | P | t-ratio | P | t-ratio | P | t-ratio | P |
| AN | 2.2 | 0.11 | 1.6 | 0.15 | 1.7 | 0.15 | -1.02 | 0.31 |
| C4 | 2.7 | 0.03 | 2.3 | 0.05 | 1.1 | .36 | 0.16 | 0.87 |
| SP | 0.22 | 0.91 | -0.16 | 0.91 | 0.12 | 0.91 | 1.7 | 0.41 |
| TNT | 1.6 | 0.25 | -0.51 | 0.61 | 1.9 | 0.24 | 1.3 | 0.25 |

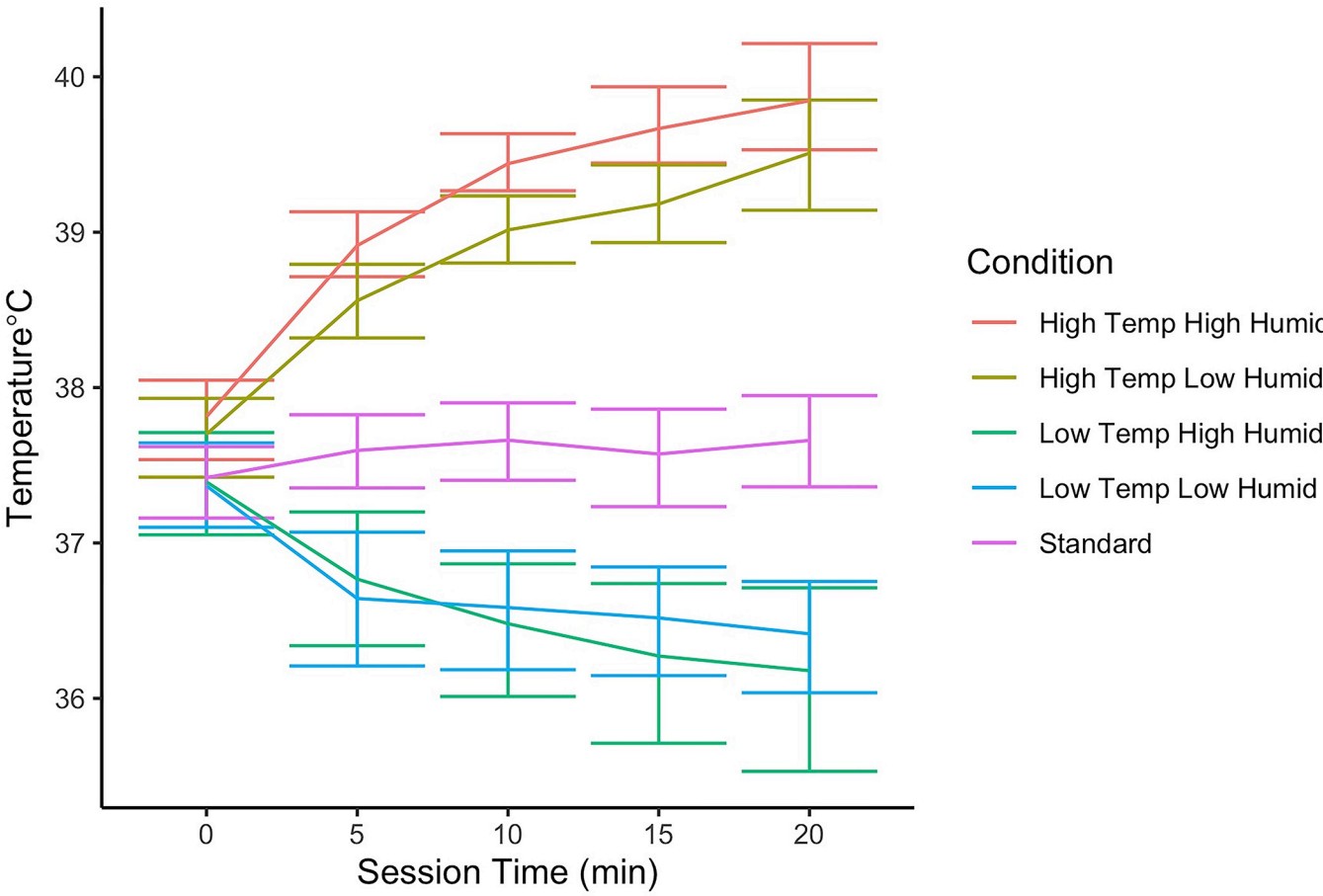

**Fig 3. Canine internal temperature change throughout a session, broken down by condition.** The maximum session time was 40m, however most dogs finished the detection task within 20 min (which is shown). Error bars show the 95% boot strap estimated confidence intervals.

condition on the temperature recorded (with a random effect of dog). Condition had a significant impact on subcutaneous temperature ($X^2$ = 1082, df = 4, p<0.0001). Post hoc tests indicate subcutaneous temperature significantly varied between all conditions except comparisons between the two low temperature conditions, which were non-significant (est = -0.071, t = -0.75, p = 0.96).

Subcutaneous temperatures in the two low temperature conditions were not accurate and as such were dropped from further analysis. The readings from the HomeAgain™ Universal Worldscan™ Reader Plus (by MERCK animal health) in low temperature were physiologically impossible for non-hypothermic dogs. These errors were likely from the chip being impacted by the temperature condition in the environmental chamber. These chips were not reading core temperature, which would have been a more accurate measure of body temperature.

Next, we evaluated the impact of mean subcutaneous temperature within a session on measured thresholds in the standard and two high temperature conditions. We conducted a mixed effect model to evaluate whether threshold was predicted by the average subcutaneous temperature across the first 25 minutes of the session (for the two hot and the standard conditions), as well as odor and their interaction. There was no significant interaction between odor and mean temperature on threshold ($X^2$ = 2.2, df = 3, p = 0.54). Threshold was however impacted by odor ($X^2$ = 122, df = 3, p<0.0001) and mean temperature ($X^2$ = 7.7, df = 1, p = 0.005). As

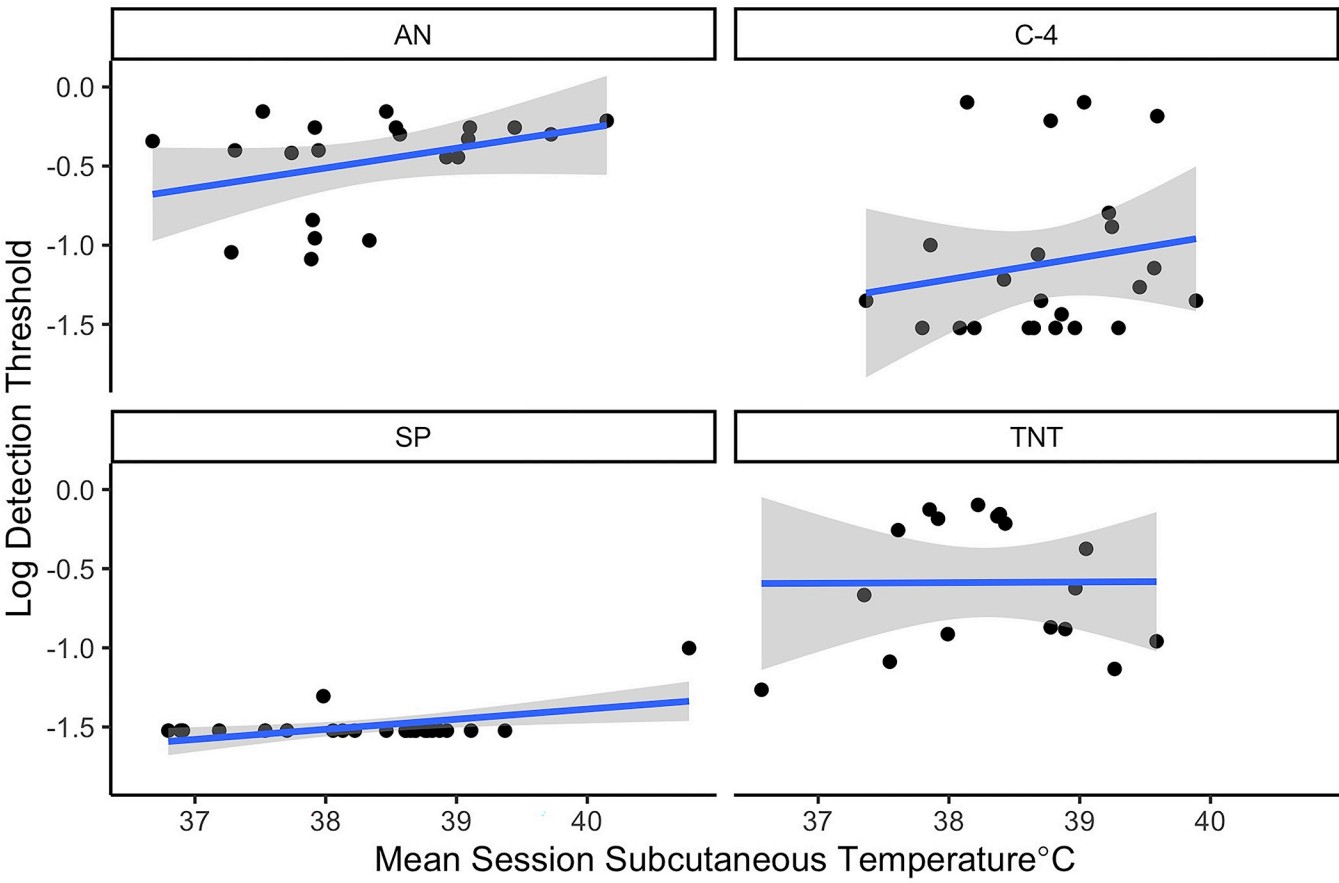

**Fig 4. Effect of Subcutaneous temperature on detection limits.** Line shows the best fit regression for each odor.

mean subcutaneous temperature in a session increased by 1˚C, threshold increased (became poorer) on average by 0.21 on the log scale. Higher mean subcutaneous temperature was associated with poorer detection limits (Fig 4). This suggests that canine subcutaneous temperature may be a simple and useful metric to predict or warn handlers of poorer detection limits.

**Latency and Inter-Olfactometer-Interval.** Two additional behavioral measures were examined for their relationship to detection threshold. First latency was measured as the time from the start of the trial until the dog inserted their nose into an odor port. Latency to start a trial did differ by the environmental conditions ($X^2 = 10.2$, p = 0.04). A post hoc test indicates, however, that the only pairwise difference between conditions was that longer latencies were observed in the low temperature high humidity condition compared to standard (t = 2.81, p = 0.04). Latency neared statistical significance between high temperature high humidity compared to standard (t = 2.64, p = 0.069).

Although there was minimal difference in latency between conditions on average, a mixed effect model was fit in which mean session latency, odor, and their interaction predicted log odor detection threshold. Results indicate there was no interaction between odor and latency ($X^2 = 1.248$, p = 0.74), but there were main effects of latency ($X^2 = 32.29$, p<0.0001) and odor ($X^2 = 169.2$, p<0.0001). For every 1s mean increase in latency there was a 0.06 increase in log threshold detection limit (poorer detection; see Fig 5).

Secondly, mean inter-box-interval was calculated for each session and trial. The inter-box-interval reflected the time the dogs took between nose pokes, reflecting time to get from one

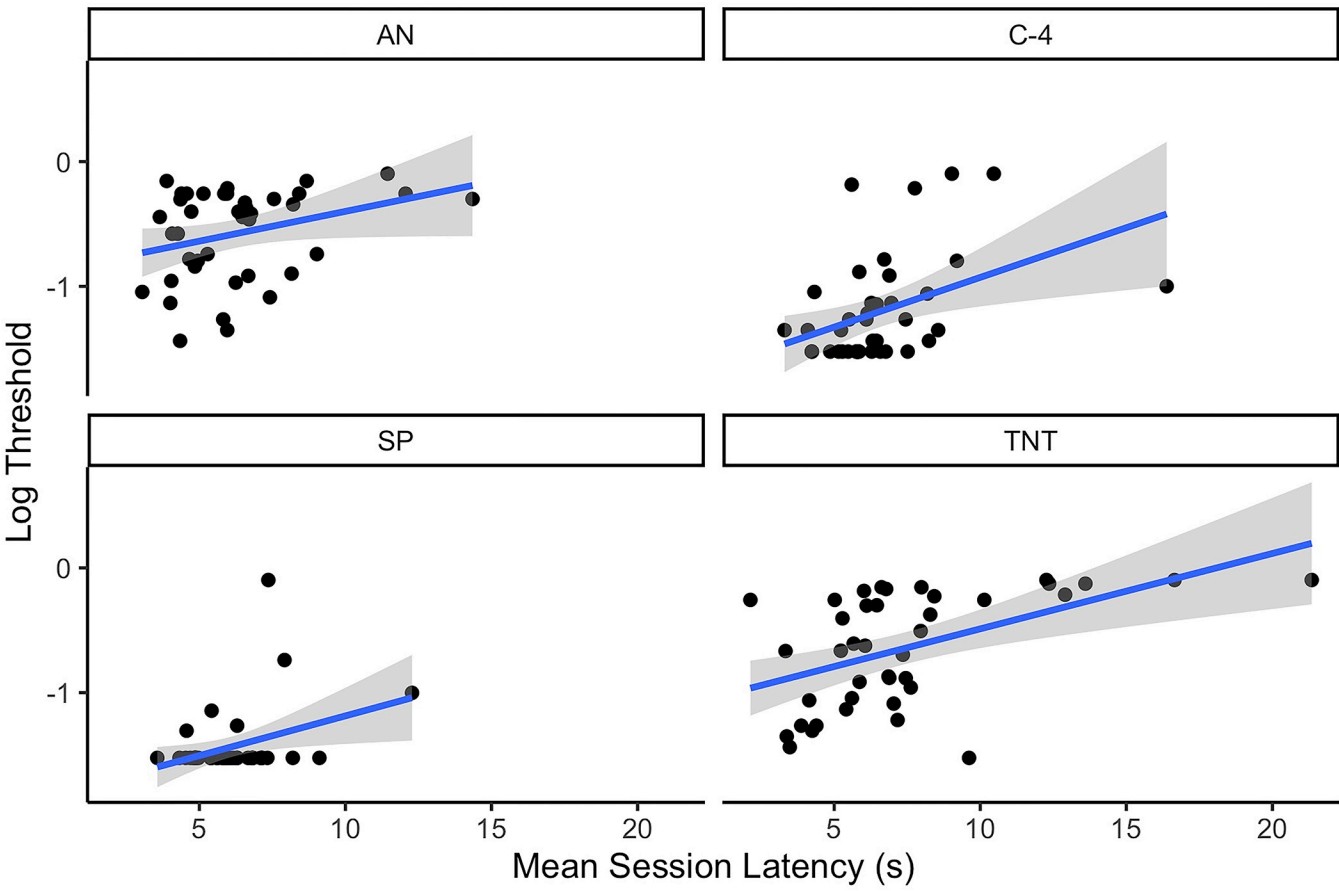

**Fig 5. Effect of mean session latency on log detection threshold.**

box to the next. The inter-box-interval did not vary in a statistically significant manner between the environmental conditions ($X^2$ = 6.57, p = 0.16) alone. Interestingly, however, the mean session inter-box-interval was related to overall threshold detection. A mixed effect model predicting threshold by the mean session inter-box-interval, odor, and their interaction, indicated that mean inter-box-interval predicted threshold ($\chi^2$ = 10.89, df = 1, p = 0.001). Odor also predicted threshold as expected ($X^2$ = 153.49, df = 3, p<0.0001) but the interaction between inter-box-interval and odor only approached statistical significance ($X^2$ = 6.87, p = 0.07). An average increase in the inter-box-interval of 1s led to an average increase in log threshold of 0.22, indicating that slower movement from one box to the next was associated with poorer threshold (see Fig 6).

**Odor availability and detection limit.** One of the goals of this experiment was to determine the interaction of odor availability and canine detection limits in adverse environmental conditions. If odor availability is a greater factor on detection limits than canine physiological response to environmental conditions, it would be expected that the best canine detection would be seen in higher temperatures and relative humidities.

Odor availability, as measured by total VOC accumulation on the SPME fiber, was measured in all environmental conditions except standard conditions. Each odor showed highest odor availability in the high temperature conditions (mean = 501 ppm) compared to the cold conditions (101 ppm) and all odors except smokeless powder showed highest odor availability

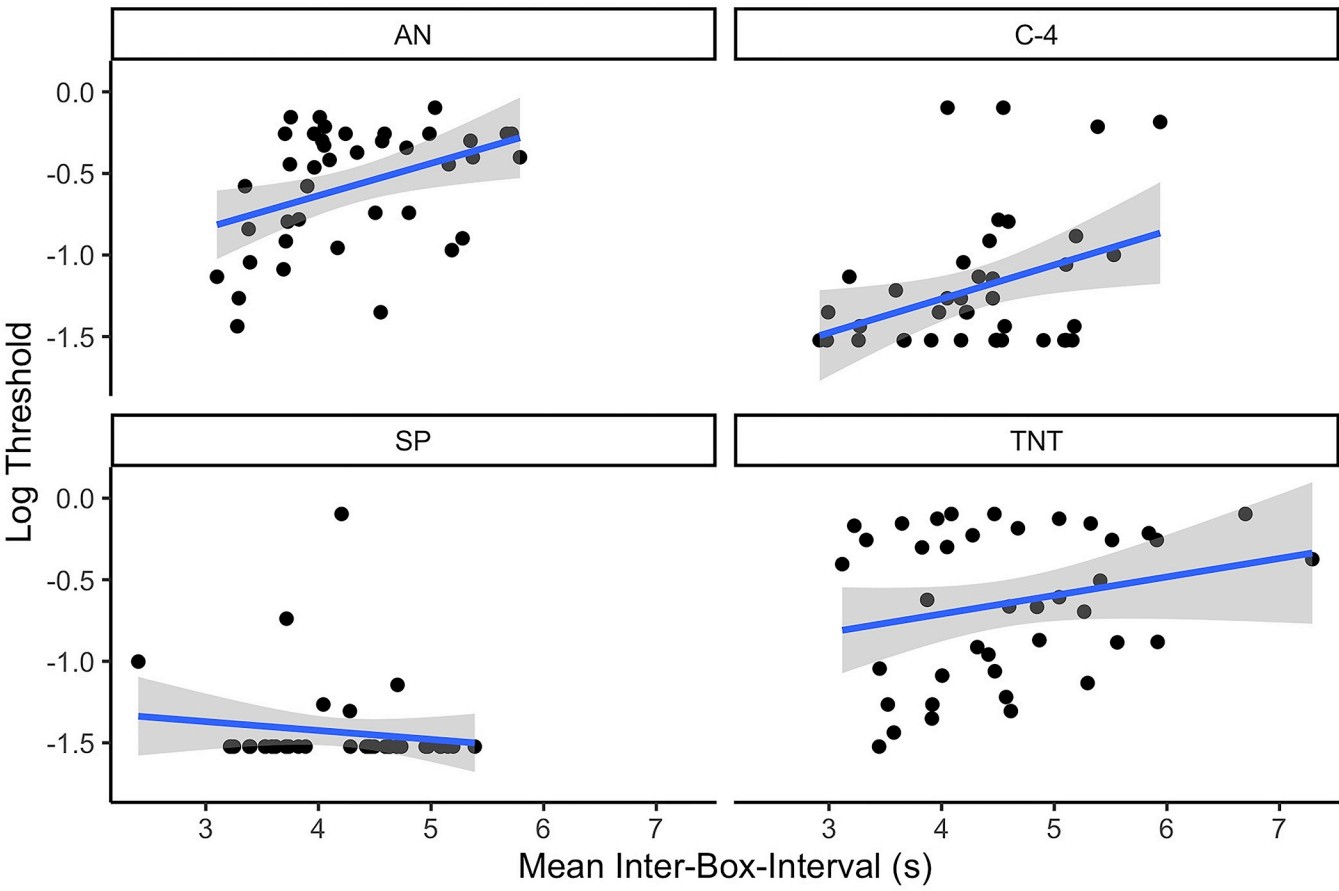

**Fig 6. Effect of the interval between olfactometer boxes on log detection threshold.**

in the high humidity conditions (mean = 334 ppm) compared to the lower humidity conditions (mean = 302 ppm).

To examine the relationship between odor availability and detection threshold, we evaluated a linear model in which detection threshold was predicted by total VOC accumulated (odor availability), with a separate model for each odor. Environmental conditions were not included due to its collinearity with VOC accumulation. Increases in VOC accumulation were associated with *poorer* detection thresholds for C-4 (estimate = 0.16, p = 0.04) and AN (estimate = 0.20, p<0.01) and was unrelated to detection threshold for TNT and SP. These results suggests that environmental conditions' impact on the canine were likely more important to detection because there was either no detected relationship between VOC accumulation and detection threshold or higher VOC accumulation/odor availability (which occurred in higher temperature and humidity conditions) led to poorer detection thresholds.

*Experiment 1 conclusion.* Overall, the results from this study suggest that decrements in detection performance appear to be largely impacted by effects on the canine. This is exemplified by performance decrements observed with C4 in high temperature conditions, where odor availability was its greatest [9]. Decreased search speed (longer inter-box-interval and search latency) and increased mean subcutaneous temperature (for the standard and high temperature conditions) were significantly associated with poorer detection limits. These behavioral and physiological measures further support that the reduced detection limits are related to canine factors rather than odor availability.

The only significant relationship seen in the data between odorant and environmental condition was between C4 and the high temperature conditions, however the results for AN are similar, although not statistically significant.

One limitation of the current study design was the small sample size (n = 8) with few within-subject replications. It could have been possible to increase the power of this experiment if each dog had been tested in each condition more than once. However, we were concerned that this would lead to acclimatization.

It was not possible to draw conclusions about threshold detection limits and environmental conditions for smokeless powder or TNT. First the dilution range of the olfactometer used was inadequate to provide the range of dilutions necessary to reach threshold for smokeless powder (lowest threshold was still above detection limit), and the highest dilution was too near the detection limit for TNT. We suspect these floor and ceiling effects could have been reduced if an olfactometer with a larger range of dilutions had been available.

An area for future study, based on Experiment 1 would be to explore more extreme environmental conditions. In particular, the cold conditions were not as extreme as what many detection dogs may need to operate in. The low temperature conditions of 0˚C was chosen based on the physical limits of the environmental chamber. Further research is necessary in subzero conditions.

The results from Experiment 1 indicate that canine detection is poorer in the high temperature conditions for C4 compared to the standard condition, However, after Experiment 1 it was not evident if this performance decrement could be reduced with specific training. Experiment 2, only studied acclimatization and C4 detection limits, because no other odor in Experiment 1 showed a significant decrement in performance compared to standard conditions. The high temperature and high humidity condition was selected as the testing and training condition because that condition resulted in significantly poorer detection compared to the standard condition.

## Experiment 2: Acclimatization training under extreme environments and canine sensitivity for an explosive odorant

### Materials and methods

**Participants.**   The same 8 dogs (4 Labrador Retrievers, 4 German Short-haired Pointers) that participated in Experiment 1 participated in Experiment 2. The dogs were assigned randomly into one of two conditions (Acclimatization and Control). These groups worked with the target odorant (C4) but were trained under different environmental acclimatization treatment plans.

**Animal welfare considerations.**   We maintained an identical criterion for early discontinuation of a session (e.g., failure to respond during trials) to that used in Experiment 1.

**Odorants.**   C4 was selected as the target odorant in Experiment 2 because it was the only target odorant which showed significant detection limit differences between the two high temperature conditions and the standard condition. All other vials in the olfactometer were empty and acted as controls, identical to the procedure in Experiment 1.

**Data collection equipment.**   Olfactometers used were identical to those used in Experiment 1.

**Environmental conditions.**   The same environmental chamber was used as in Experiment 1. However, only two of the environmental conditions were utilized, the standard conditions (22˚C and 50% relative humidity), and high temperature, high humidity (40˚C and 70% relative humidity). In both conditions there was a 3˚C and 5% relative humidity window on either side to allow changes caused by the chamber door opening and closing.

The average high outdoor temperature during the duration of this experiment was 37.8˚C with lows of 23.1˚C. Dogs were kept in runs with climate control and outdoor access. Dogs were walked twice daily for the duration of this study. Dogs were not intentionally acclimated to this outdoor temperature but had outdoor access for welfare purposes.

**Canine acclimatization training and testing.** The eight dogs were randomly split into two groups: control and acclimatization. This threshold test followed identical procedures to that of Experiment 1 with the exception of completing four reversals instead of seven. Both groups received equal amounts of threshold training.

Dogs in the experimental group received 5-minute exposure with play-based exercise within the environmental condition followed by a threshold detection session. The purpose of this pre-odor task exercise was to work the dogs while they were fatigued, which Gazit and Terkel (2003), found decreased detection limits initially. Exercise treatment consisted of either a toy toss (Kong™) or food toss (the food used as their reward during odor training and testing) prior to the odor threshold assessment. This five-minute window was chosen because the dogs were panting after 5 minutes, but still willing and able to work without laying down. Prior to the start of the experiment, a preference test was performed to determine which reward the dog would prefer to work for in this initial 5 minutes of exercise. In this test, the experimenter tossed a toy and a piece of the dog's normal kibble out in front of the restrained dog. The experimenter then released the dog and whichever reward the dog engaged with first was recorded. This procedure was repeated five times, and whichever reward the dog chose more was the reward that was used in acclimatization training. Three of the four dogs initially used the Kong™ as their reward. However, after the dogs reached the high temperature and high humidity condition requirements (Day 7) only 1 dog still engaged with the toy. The other two dogs were switched to a food toss, which they continued to engage with the rest of the study. The environmental conditions for the acclimatization group, slowly escalated over the course of six days to reach the high temperature and high humidity conditions (see Table 7).

In contrast, the control group received a five-minute pet session in the environmental chamber (instead of exercise) followed by a threshold detection session. Critically, environmental conditions remained at standard conditions during the training phase for control dogs and control and experimental dogs received identical olfactory training.

**Canine testing.** To ensure that the two experimental groups did not have significantly different detection limits for C4 by random chance, on Day 0, the eight dogs were tested (with C4) in the standard temperature and RH condition, without prior exercise.

On day 11 (after 10 days of acclimatization training) and on day 22 (after twenty days of acclimatization training) both groups underwent threshold testing. This testing consisted of the same threshold test done during acclimatization, in the absence of the exercise/pet five-minute prior and completed at the high temperature and high humidity condition (40˚C and 70% RH).

**Table 7. Details the environmental conditions in the acclimatization plan by group.** All temperatures are in ˚C.

| Day | Acclimatization group (n = 4) condition | Control group (n = 4) condition |
| --- | --- | --- |
| 1 | 21˚ ± 3˚; 50% ± 5% | 21˚ ± 3˚; 50% ± 5% |
| 2 | 24˚ ± 3˚; 53% ± 5% | 21˚ ± 3˚; 50% ± 5% |
| 3 | 28˚ ± 3˚; 56% ± 5% | 21˚ ± 3˚; 50% ± 5% |
| 4 | 31˚ ± 3˚; 59% ± 5% | 21˚ ± 3˚; 50% ± 5% |
| 5 | 34˚ ± 3˚; 62% ± 5% | 21˚ ± 3˚; 50% ± 5% |
| 6 | 38˚ ± 3˚; 65% ± 5% | 21˚ ± 3˚; 50% ± 5% |
| 7–22 | 40˚ ± 3˚; 70% ± 5% | 21˚ ± 3˚; 50% ± 5% (excluding testing days 10 and 22) |

Both testing and acclimatization training sessions lasted approximately 30 minutes, including the pre-training five minutes treatment. These sessions ended if the dog met welfare criterion, had been in the chamber for 40 minutes without achieving four reversals, achieved four reversals, or got three correct responses at the lowest concentration.

**Data analysis.** Similar to Experiment 1 threshold was calculated as the geometric mean of all four reversal points. Odor threshold data was log transformed for all analyses. Threshold limits were imputed through the same calculations as done in Experiment 1. If dogs' detection exceeded the maximum dilution capabilities, the lowest dilution point was used as a reversal point for the remaining reversal points. If a dog did not complete the threshold assessment due to meeting a welfare criterion, prior to reaching all four reversal points, the remaining reversal points were imputed as 80%. Additionally, we also analyzed the results without imputing missing threshold values and calculated threshold using the available reversal points prior to the end of the session.

The effect of treatment (group) on detection threshold was evaluated using a linear mixed model effect, where threshold was predicted by treatment group with a random effect of dog. The model included the threshold data from both the first test (following 10 days acclimation) and the second test (following 20 days acclimation). Additional models were fit to evaluate how treatment group impacted the dependent variables of subcutaneous temperature, inter-box interval and search latency.

## Results and discussion

Fig 7A shows that there were minimal differences in detection limit at Day 0 under standard conditions. In fact, detection limits were slightly poorer for the acclimation group compared to the control. Fig 7B, however, shows that acclimation dogs outperformed control dogs under the high temperature and high humidity conditions averaged across Test 1 and Test 2 with 10 and 20 days of acclimatization training. Fig 7C shows that acclimation dogs showed optimal threshold performance following just 10 days of acclimatization training, outperforming control dogs. A similar result is seen for Test 2, although with less variability in the control dogs and slightly more variability for the acclimated dogs.

A linear mixed effect model predicting detection threshold by treatment across Test 1 and Test 2 indicated that acclimatization training had a trend effect when using the imputed thresholds ($X^2 = 2.9$, df = 1, p = 0.09) and a statistically significant effect when not imputing missing thresholds ($X^2 = 4.2$, df = 1, p = 0.041). Only one of the eight dogs was removed from the testing room during a testing day (Test 1) due to the welfare criterion. This dog was in the control group.

Fig 8 shows the effect of acclimatization training on (control vs. acclimatization) threshold (imputed values) and for the three measures that were previously found to be associated with detection thresholds in Experiment 1 (mean subcutaneous temperature, mean inter box interval, and mean session latency). In each of the following models temperature, mean session latency, and mean inter box interval were compared between the two treatment groups for the two test days (excluding day 0, which was at standard conditions). A linear mixed effect model indicates that mean temperature was not different by treatment across the two test days at high temperature and high humidity ($X^2 < 0.1$ df = 1, p = 0.98). In contrast, there was a significant difference between treatments on the mean inter-box-interval, such that acclimatization group walked on average 1s faster from one box to the next than control dogs during the test days ($X^2 = 9.66$, df = 1, p = 0.002). In addition, latency almost met the criterion for statistical significance, and showed that acclimatization dogs-initiated trials approximately 3s faster than did controls during the high temperature high humidity tests ($X^2 = 3.35$, df = 1, p = 0.07).

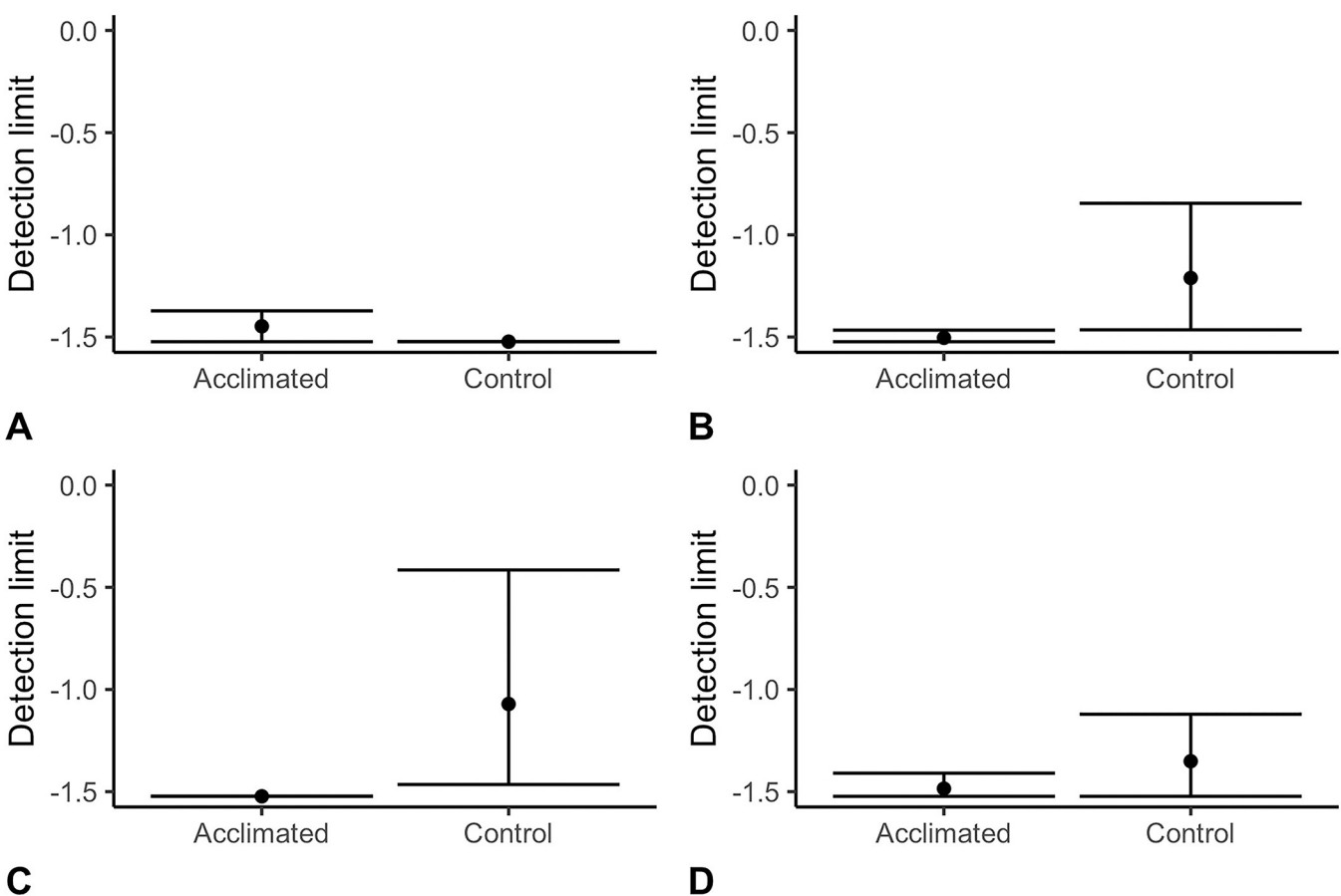

**Fig 7. Detection limits for each of the two treatment groups on a baseline standard condition.** A) initial test day, B) average of Test 1 and 2, C) Test 1, and D) Test 2 at 20 days. This graph used imputed detection values.

Interestingly, subcutaneous temperature was not different between the treatment and control group. This suggests that although behavioral performance improved (e.g., inter-box interval) and detection limits improved, this was not likely directly caused by improved physiological responses to heat stress (e.g., subcutaneous temperature).

The acclimatization protocol leveraged for this study combined brief exercise with de-sensitization and exposure training to the environmental condition. Although this combined program (exercise + de-sensitization) led to improved performance, it's not clear if both components are necessary. However, given that the exercise conditioning was minimal (~5 minutes) it is possible that a more extensive fitness training program [15, 16] could lead to greater benefits.

Overall, results from the acclimatization treatment show that brief exercise and de-sensitization to working in 40˚ ± 3˚; 70% ± 5% mitigated detection threshold decrements for C4 in those environmental conditions. Performance improvements were also observed for mean inter-box interval (search speed). Further, performance improvements were observed after 10 days of conditioning, and we did not observe any greater performance gains following 20 days of acclimatization training compared to 10. Future research is needed to investigate separate effects of physical conditioning and de-sensitization to the environmental conditions to identify optimal and sufficient combinations to mitigate performance decrements.

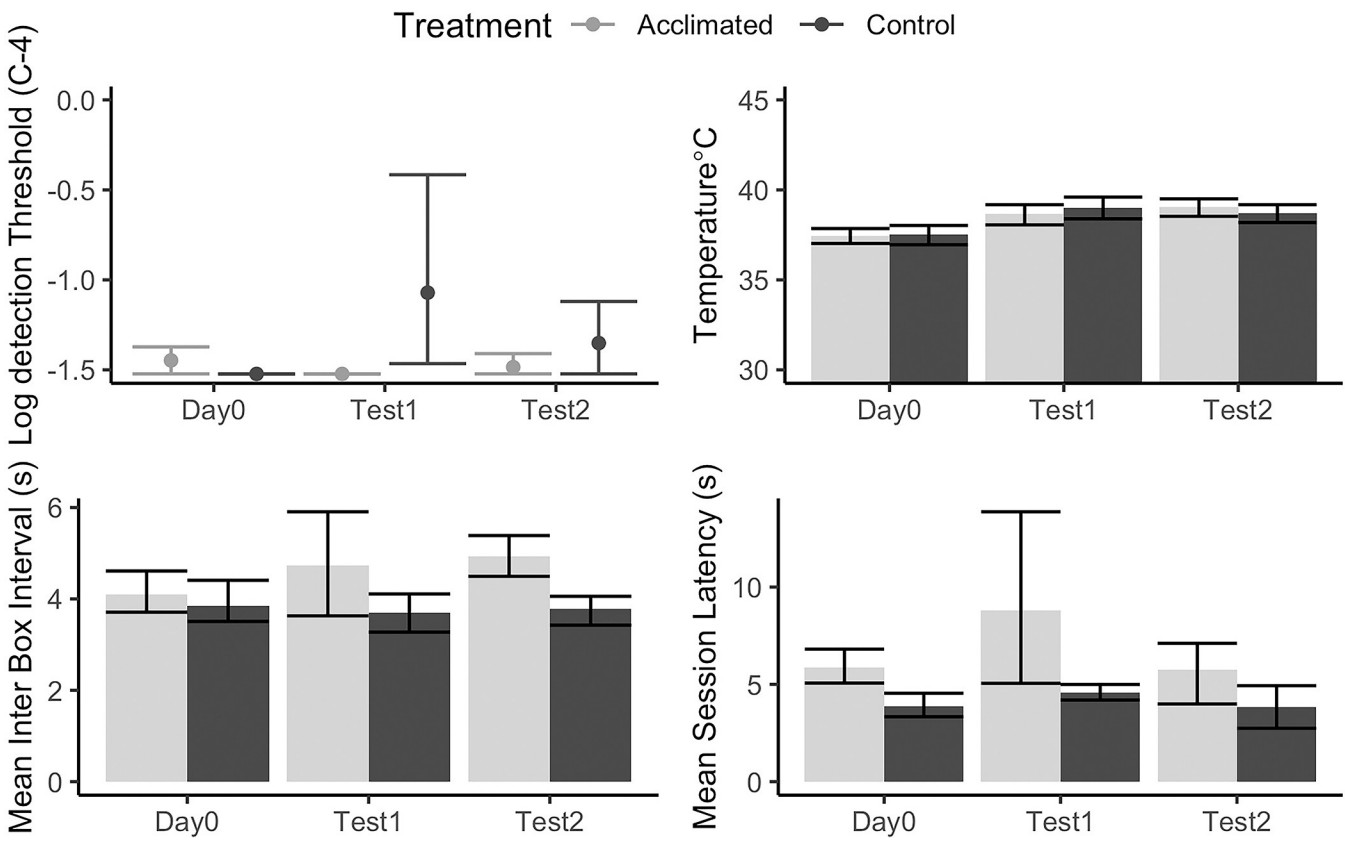

**Fig 8. Effect of acclimatization on detection threshold, subcutaneous temperature, inter box interval and latency.** Day 0 is the baseline at standard conditions. Test 1 and Test 2 is under High Temperature and High Humidity conditions following 10 and 20 days of acclimatization, respectively. All threshold values were based on imputed calculations.

## General discussion

The goal of this study was to determine how canine detection limits for four explosive odorants change with varying environmental conditions, considering both canine factors and odor availability (Experiment 1). An acclimatization plan was also developed to assess how exercise and environmental exposure affect canine detection limits in adverse environmental conditions (Experiment 2).

The results from Experiment 1 indicated that canine detection limits for C4 in the 21°C 70% relative humidity condition was 3.5-fold poorer than at standard conditions. These results suggest that canine physiological factors have a greater impact on detection performance than odor availability, because odor availability was higher in this condition compared to standard [9].

Canine subcutaneous temperatures were significantly higher in the 40°C 70% relative humidity condition compared to the 40°C 40% relative humidity condition indicating that humidity does play a role in canine subcutaneous temperature. Likely this is because evaporative cooling is less effective in higher humidity, and panting is one of the main ways canines cool [6]. This finding contradicts results from racing greyhounds, in which relative humidity was not a significant factor in canine rectal temperature post-race, but ambient temperature alone was [17]. However, this study did note that in the climates studied temperature and relative humidity were inversely related. McNicholl et al., (2016) were not able to study the impact of high temperatures and differing levels of relative humidity on canine temperature.

Even within high temperature conditions, higher mean subcutaneous temperatures were significantly related to poorer detection limits. These results support the findings of Gazit and Terkel (2003) which found that dogs with a higher panting ratio had more difficulty detecting explosives. Additionally, this finding supports the results of Brustkern et al., 2023, which showed that the largest performance decrement was in the high temperature high humidity condition. This suggests that methods which reduce the canine body temperature could possibly mitigate declines in detection limits.

Previous studies indicate that some cooling methods work more quickly than others, and therefore are likely preferable for EDC teams working a long search [18]. Davis et al. (2019) found that cooling via immersion was faster for cooling gastrointestinal temperatures than by either cold mats or fans. This effect was even true if the water used to immerse the dog was not cool. Water immersion reduces the internal body temperature fastest, and thus may reduce the panting to sniffing ratio the quickest and thus allow EDC teams to return to work more quickly.

In addition to changes in subcutaneous temperature, canines that showed increased latency to search and increased inter-box-interval, had poorer detection performance in Experiment 1. Changes in search speed may provide a valuable resource for handlers to monitor canine performance. Previous research, which studied a population of scent detection canines, found that total activity during three tasks: agility, toy retrievals and searching decreased with increasing outdoor temperature [19]. While this study did not investigate how search accuracy related to ambient temperatures, it does provide support that temperature effects activity.

In Experiment 2 canine detection performance increased for C4 in the 40˚C 70% relative humidity condition after 10 days of acclimatization training (training in increased temperatures and exercise) compared to non-acclimated dogs. There were no changes in subcutaneous temperature between groups suggesting there were no physiological temperature changes resulting from the acclimatization plan. However, canines in the acclimatization group did have increased search speed (reduced latency and inter-box-interval). It is unclear from Experiment 2 if the exercise or the increased exposure to adverse environmental conditions or both are necessary for increased detection performance, further research is needed to clarify this relationship.

One important limitation for this study is the limited sample size. Nonetheless, in Experiments 1 and 2, clear differences in performance were observed. A second limitation was the dilution capabilities of our olfactometer system. This limited the ability to measure differences between groups because thresholds for some odors reached a ceiling or floor. Future work could utilize multi-stage dilution to generate higher and lower dilutions, which would be more sensitive to detecting odor threshold changes.

## Future directions

Future work that refines an acclimatization plan would be valuable. Experiment 2 indicated that after just 10 days of increased exposure to adverse environmental conditions with five minutes of exercise canines had better detection for C4 than their counterparts who did not work in high temperature and humidity conditions and were not exercised prior to work. However, it is not clear, based on this research if both exercise and increased exposure to adverse conditions are necessary and whether it is necessary to acclimate dogs for 10 days for better detection limits, or if fewer days could produce the same results, and thus reduce training time.

Lastly, our analysis was focused on the decrements caused by high temperatures on C4 detection limits. However, our prior work in Experiment 1 demonstrated that canines can also

show reduced performance in cold temperatures, particularly cold and humid conditions. Thus, additional work is necessary to explore de-sensitization and/or physical conditioning effects on performance decrements in cold conditions.

## Conclusions

Poorer detection limits for C4 were observed in high temperatures (both at high and low humidity) conditions compared to standard conditions. Similar trends were observed with other energetics, but were less clear due to limitations in odor dilutions our olfactometer system generated. Poorest performance for C4 detection occurred under highest temperature and humidity conditions in which there was highest VOC availability as measured through SPME GC-MS, indicating that performance decrement was likely due to canine factors rather than odor availability In Experiment 2, C4 detection limits improved in the high temperature high humidity condition following an acclimatization plan compared to control dogs within 10 days. Improvements were associated with behavioral (inter-box interval) changes rather than changes in subcutaneous temperature.

## Supporting information

**S1 Dataset. Contains raw olfactometer data from Experiment 1 of the current paper.** (CSV)

**S2 Dataset. Contains raw olfactometer data from Experiment 2 of the current paper.** (CSV)

**S3 Dataset. Contain the same subcutaneous temperature data from Experiment 1, in two formats.** (ZIP)

**S4 Dataset. Contain subcutaneous temperature data from Experiment 2 in two formats.** (ZIP)

**S1 File. The R code (QMD) used for analysis and to clean the data prior to analysis for Experiments 1 and 2 can be found in these files: "StageIIAnalysis_Plos1.qmd" and "Stage-IIIAnalysis_Plos1.qmd".** (ZIP)

**S1 Data. Contains the data from the analytical chemistry analysis, used to run the odor availability analysis in Experiment 1.** (CSV)

## Acknowledgments

The authors would like to thank all husbandry staff, and volunteers at the Canine Olfactory Education and Research lab at Texas Tech University for their care of the canines during this project. Special thanks to Avery Bramlett for her help on Experiment 2.

## Author Contributions

**Conceptualization:** Paola A. Prada-Tiedemann, Nathaniel J. Hall.

**Data curation:** Sarah A. Kane, Dillon E. Huff, Paola A. Prada-Tiedemann, Nathaniel J. Hall.

**Formal analysis:** Sarah A. Kane, Dillon E. Huff, Paola A. Prada-Tiedemann, Nathaniel J. Hall.

**Funding acquisition:** Paola A. Prada-Tiedemann, Nathaniel J. Hall.

**Investigation:** Sarah A. Kane, Lauren S. Fernandez, Dillon E. Huff.

**Methodology:** Sarah A. Kane, Dillon E. Huff, Paola A. Prada-Tiedemann, Nathaniel J. Hall.

**Project administration:** Paola A. Prada-Tiedemann, Nathaniel J. Hall.

**Resources:** Dillon E. Huff, Paola A. Prada-Tiedemann, Nathaniel J. Hall.

**Software:** Nathaniel J. Hall.

**Supervision:** Nathaniel J. Hall.

**Validation:** Dillon E. Huff, Paola A. Prada-Tiedemann.

**Visualization:** Sarah A. Kane, Nathaniel J. Hall.

**Writing – original draft:** Sarah A. Kane.

**Writing – review & editing:** Lauren S. Fernandez, Dillon E. Huff, Paola A. Prada-Tiedemann, Nathaniel J. Hall.

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
