## [Decision Letter · Decision Letter 0]

8 Nov 2023

PONE-D-23-31666Canine detection of explosive odorants in extreme environmental conditions and development of an acclimatization plan to improve sensitivityPLOS ONE

Dear Dr. Kane,

Thank you for submitting your manuscript to PLOS ONE. After careful consideration, we feel that it has merit but does not fully meet PLOS ONE’s publication criteria as it currently stands. Therefore, we invite you to submit a revised version of the manuscript that addresses the points raised during the review process.

We look forward to receiving your revised manuscript.

Kind regards,

Rajeev Singh

Academic Editor

PLOS ONE

 [This research was made possible through funding provided by the DoD Army Research Office under Contract No. W911NF2120124. The views expressed in this manuscript are those of the author and do not necessarily reflect the official policy or position of the Department of Defense, nor the U.S. Government. https://www.arl.army.mil/who-we-are/aro/

SAK’s work was supported by the National Science Foundation Graduate Research Fellowship Program (DGE 2140745). Any opinions, findings, conclusions or recommendations expressed in this work are those of the author(s) and do not necessarily reflect the views of the National Science Foundation.https://www.nsf.gov/].  

Reviewers' comments:

Reviewer's Responses to Questions

**Comments to the Author**

1. Is the manuscript technically sound, and do the data support the conclusions?

Reviewer #1: Yes

Reviewer #2: Yes

Reviewer #3: Yes

2. Has the statistical analysis been performed appropriately and rigorously? 

Reviewer #1: Yes

Reviewer #2: Yes

Reviewer #3: Yes

3. Have the authors made all data underlying the findings in their manuscript fully available?

Reviewer #1: Yes

Reviewer #2: Yes

Reviewer #3: Yes

4. Is the manuscript presented in an intelligible fashion and written in standard English?

Reviewer #1: Yes

Reviewer #2: Yes

Reviewer #3: Yes

5. Review Comments to the Author

Reviewer #1: There is a minor correction needed, please check the highlighted area in the text. However, there are one or two other areas to look for and give a one more reading to make an improvement in writing.

Reviewer #2: This manuscript presents some important results related to the detection of explçosives by dogs and the effect of ambient temperature/relative humidity on this skill.

However, some information is still required to understand the results.

How long did the dogs remain inside the climatic chamber at each temperature?

How was the outside temperature at which they were acclimatized? It is important to show this information in both experiments.

Is there a table/figure citation on line 335?

This symbol Ꭓ2 does not appear in the manuscript.

Reviewer #3: Title : Superb effort has been made by authors. Make title eye-catchy for the readers. Make it distinct from other research papers in the same field.

Abstract: Make initial impression to the point. Start from main aim of current investigation. Write a concluding line reflecting future direction.

Introduction: Highlight the canine and odor interactive effects. improve introduction section with latest references.

General Discussion : This section is deficient in supported studies. Add previous investigations in line with current findings.

If possible add latest refrences

6. PLOS authors have the option to publish the peer review history of their article (what does this mean?). If published, this will include your full peer review and any attached files.

Reviewer #1: **Yes: **Muhammad Tariq Javed

Reviewer #2: **Yes: **Cristiane Gonçalves Titto

Reviewer #3: No

---

## [Author Response · Author response to Decision Letter 0]

29 Dec 2023

Kane Plos1 rebuttal letter

Reviewer #1: 

There is a minor correction needed, please check the highlighted area in the text. However, there are one or two other areas to look for and give a one more reading to make an improvement in writing. 

Thank you, we have reviewed the manuscript and have made several revisions to improve the writing. Special attention was paid to the highlighted sections. 

Reviewer #2: 

This manuscript presents some important results related to the detection of explosives by dogs and the effect of ambient temperature/relative humidity on this skill.However, some information is still required to understand the results.

How long did the dogs remain inside the climatic chamber at each temperature? 

Thank you for noting this. This detail has been added, lines 285-289 (experiment 1); 509-512 (experiment 2). 

How was the outside temperature at which they were acclimatized? It is important to show this information in both experiments. 

This has been added for experiment 1 in lines 233-237. Dogs were not intentionally acclimated in outdoor temperatures during this study; however, they did have outdoor access and each dog was walked 2x per day. For experiment 2 this was addressed in lines 476-479. 

Is there a table/figure citation on line 335?

 Yes, we have re-inserted figure 3 (included in submission as Fig3.tif).

This symbol Ꭓ2 does not appear in the manuscript. 

Thank you for the comment, all have been changed to X2

Reviewer #3: Title : 

Superb effort has been made by authors. 

Thank you! 

Make title eye-catchy for the readers. Make it distinct from other research papers in the same field.

I added a (hopefully) catchy first part to the title. 

Abstract: 

Make initial impression to the point. Start from main aim of current investigation. 

Thank you for your suggestion, we have changed the line to read “Canines are one of the best biological detectors of energetic materials available; however, canine detection of explosives is impacted by a number of factors, including environmental effects.”

Write a concluding line reflecting future direction. 

We have added this final sentence to the abstract: Future work would endeavor to determine what aspect (exercise or environmental conditioning) are most effective in acclimatization for odor detection work.

Introduction: 

Highlight the canine and odor interactive effects. 

Added lines 128-130 discussing potential interactive factors in higher temperatures. 

Improve introduction section with latest references.

We have re-reviewed the literature, using search terms “canine”, “odor detection” and “environment” and all of the relevant literature is included in the introduction 

General Discussion: 

This section is deficient in supported studies. Add previous investigations in line with current findings.

If possible add latest references.

Thank you for this note, several studies have been added to our general discussion to help ground our results in current literature.

---

## [Editor Report · Decision Letter 1]

9 Jan 2024

Canine Detection of Explosives Under Adverse Environmental Conditions With and Without Acclimation Training

PONE-D-23-31666R1

Dear Dr. Kane,

We’re pleased to inform you that your manuscript has been judged scientifically suitable for publication and will be formally accepted for publication once it meets all outstanding technical requirements.

Kind regards,

Rajeev Singh

Academic Editor

PLOS ONE
---

## [Editor Report · Acceptance letter]

31 Jan 2024

PONE-D-23-31666R1 

PLOS ONE

Dear Dr. Kane, 

I'm pleased to inform you that your manuscript has been deemed suitable for publication in PLOS ONE. Congratulations! Your manuscript is now being handed over to our production team.

Kind regards, 

on behalf of

Dr. Rajeev Singh 

Academic Editor

PLOS ONE